# The Effect of High Protein Powder Structure on Hydration, Glass Transition, Water Sorption, and Thermomechanical Properties

**DOI:** 10.3390/foods11030292

**Published:** 2022-01-21

**Authors:** Valentyn A. Maidannyk, David J. McSweeney, Sharon Montgomery, Valeria L. Cenini, Barry M. G. O’Hagan, Lucille Gallagher, Song Miao, Noel A. McCarthy

**Affiliations:** 1Food Chemistry and Technology Department, Teagasc Food Research Centre, Moorepark, Fermoy, P61 C996 Co. Cork, Ireland; Valentyn.maidannyk@teagasc.ie (V.A.M.); David.mcsweeney@teagasc.ie (D.J.M.); Sharon.montgomery@teagasc.ie (S.M.); Song.Miao@teagasc.ie (S.M.); 2School of Food and Nutritional Sciences, University College Cork, T12 K8AF Co. Cork, Ireland; 3School of Biomedical Sciences, University of Ulster, Coleraine BT52 1SA, UK; v.cenini@ulster.ac.uk (V.L.C.); bmg.ohagan@ulster.ac.uk (B.M.G.O.); Gallagher-L31@ulster.ac.uk (L.G.)

**Keywords:** gas injection, milk protein concentrate (MPC), glass transition, dynamic mechanical analysis (DMA), α-relaxation, structural strength, environmental scanning electron microscope (ESEM)

## Abstract

Poor solubility of high protein milk powders can be an issue during the production of nutritional formulations, as well as for end-users. One possible way to improve powder solubility is through the creation of vacuoles and pores in the particle structure using high pressure gas injection during spray drying. The aim of this study was to determine whether changes in particle morphology effect physical properties, such as hydration, water sorption, structural strength, glass transition temperature, and α-relaxation temperatures. Four milk protein concentrate powders (MPC, 80%, *w*/*w*, protein) were produced, i.e., regular (R) and agglomerated (A) without nitrogen injection and regular (RN) and agglomerated (AN) with nitrogen injection. Electron microscopy confirmed that nitrogen injection increased powder particles’ sphericity and created fractured structures with pores in both regular and agglomerated systems. Environmental scanning electron microscopy (ESEM) showed that nitrogen injection enhanced the moisture uptake and solubility properties of RN and AN as compared with non-nitrogen-injected powders (R and A). In particular, at the final swelling at over 100% relative humidity (RH), R, A, AN, and RN powders showed an increase in particle size of 25, 20, 40, and 97% respectively. The injection of nitrogen gas (NI) did not influence calorimetric glass transition temperature (T_g_), which could be expected as there was no change to the powder composition, however, the agglomeration of powders did effect T_g_. Interestingly, the creation of porous powder particles by NI did alter the α-relaxation temperatures (up to ~16 °C difference between R and AN powders at 44% RH) and the structural strength (up to ~11 °C difference between R and AN powders at 44% RH). The results of this study provide an in-depth understanding of the changes in the morphology and physical-mechanical properties of nitrogen gas-injected MPC powders.

## 1. Introduction

Poor hydration and solubility are characteristics of high protein milk powder reconstitution, which are critical attributes for manufacturers and end-users. Low solubility and hydration properties mean that a substantial amount of powder remains undissolved after reconstitution in water. However, there have been a significant number of technological advancements to improve high protein powder solubility. Modifying chemical composition (proteins, lactose, minerals, etc.), physical properties (initial and intermediate water content of powders, temperature, and time of reconstitution), and the use of technological processes such as homogenization, cavitation, and ultrasonication have been shown to influence final powder solubility [1,2,3,4,5,6,7,8].

Recently, gas injection during spray drying has been shown to improve the solubility of high protein powders by creating vacuoles and pores, which allow for rapid water uptake and transfer into the particles. This study is one of a series of publications on the effects of nitrogen (N_2_) gas injection prior to spray drying that examines the physical properties of milk protein concentrate (MPC) powders, mainly focused on hydration, phase transition, powder structural changes, and stability as a function of temperature, time, and humidity. Previous studies have shown that high-pressure gas injection (e.g., carbon dioxide, compressed air, and nitrogen) can significantly improve the quality attributes of several different types of milk products, such as dairy powders and yogurts [9]. Gas can be introduced at different stages during processing, for example, during membrane filtration, after evaporation, or during spray drying, and can influence the powder physicochemical properties, such as particle, morphology, bulk density, solubility [10], and dispersibility [11,12]. Earlier, Hanrahan et al. (1962) and Bell et al. (1963) showed that compressed air or N_2_ injection incorporated after the high-pressure pump and before the nozzles, improved skim milk powder dispersibility [13,14]. McSweeney et al. (2021a) showed that the injection of N_2_ into MPC liquid concentrate prior to spray drying significantly improved powder solubility, but also found that agglomeration of MPC powders resulted in a significant reduction in solubility and particle dissociation, indicating that particle structure plays a major role in powder solubility [15]. Overall, the improvement in solubility has been attributed to the newly created air voids and vacuoles in the powder particles as a result of the release of air from inside the droplets during atomization. However, it is yet to be elucidated how particle morphology affects the thermodynamic and kinetic properties of compositionally similar powders.

Water added to amorphous systems such as anhydrous milk powders behaves as a crystallizing solvent. Typically, the presence of water dramatically changes the powder physical properties such as the calorimetric glass transition temperature (T_g_) of amorphous systems [16]. The structural strength (S) concept [17,18,19,20] based on the Williams–Landel–Ferry (WLF) equation sufficiently explains the factors responsible for the molecular mobility around and below T_g_ that characterize the differences in powder solubility [21]. This approach includes water sorption isotherms, differential scanning calorimetry (DSC) to measure T_g_, dynamic mechanical analysis (DMA) to measure α-relaxation temperature (T_α_) at various relative humidities (RHs), and rheology for high-water content systems. This analysis has been successfully applied to various food and dairy systems including complex semi-crystalline and encapsulated systems [18,19,20,22,23,24,25].

Microscopy is a powerful tool which facilitates individual dairy powder particle imaging at a micro and nanoscale [26]. Recently, an environmental scanning electron microscope was employed to observe changes in the surface microstructure of MPC powders spray dried in situ under different RH conditions [27]. This technique allows the visualization at high magnification and spatial resolution of the hydration and reconstitution process of individual dairy powder particles in real time. It also permits characterization of the resulting residue (i.e., dissolved or undissolved particles) left after rehydration.

The aim of this study was to determine the influence of N_2_ injection and agglomeration on the physicochemical properties and morphology/structure of MPC powders, all of which have the same chemical composition.

## 2. Materials and Methods

### 2.1. Materials and Manufacture of Milk Protein Concentrate Powders

Four milk protein concentrate powders (MPC, 80%, *w*/*w*, protein) were produced using the pilot-plant facilities at Moorepark Technology Limited (Teagasc, Moorepark, Fermoy, Co Cork, Ireland), i.e., regular (R) and agglomerated (A) without N_2_ injection, and regular (RN) and agglomerated (AN) with N_2_ injection, as previously described in detail by McSweeney et al. (2021a) [15]. Briefly, milk protein concentrate (MPC) powder (80%, *w*/*w*, protein) was supplied by a local dairy ingredient manufacturer and rehydrated using an in-line Crepaco high shear mixer (APV Pulvermixer, SPX Flow Technology, Pasteursvej, Silkeborg, Denmark) to produce an MPC dispersion (21.2%, *w*/*w*, total solids, 2370 kg of product). Then, the dispersion was passed once through an SPX hydrodynamic cavitator (Model P286184-12 R4, SPX Flow Technology, Pasteursvej, Silkeborg, Denmark) to ensure full rehydration of the powder. The MPC dispersion was subsequently heated to 70 °C using a scraped surface heat exchanger before being pumped to the atomization nozzles using a high-pressure pump (HPP). Nitrogen (N_2_) gas was injected (3.5 kg/h) at a pressure of ~190 bar into the feed line, after the HPP and prior to atomization, using a pressurized injection unit (Carlisle Process Systems, Farum, Denmark). Drying was performed using a NIRO Tall Form spray dryer (TFD-0025-N, Soeborg, Denmark), with air inlet and outlet temperatures set at 185 °C and 75 °C, respectively, for the production of regular and agglomerated powders. Air inlet and outlet temperatures were set at 180 °C and 75 °C, respectively, for both N_2_-injected regular and agglomerated powders. For agglomeration, fines were transferred from the cyclone to the top of the spray dryer and introduced at close proimity to the atomization nozzles. For regular powders, all fines were returned to the external fluid bed.

### 2.2. Environmental Scanning Electron Microscopy

All environmental scanning electron microscopy experimental and analysis work was performed at Ulster University’s Bio-Imaging Core Facility Unit (Coleraine, Northern Ireland). The topography and structural alterations of all MPC powder particles during the experiments were observed with a FEI Quanta^TM^ 200 (FEI Company, Eindhoven, The Netherlands) Environmental scanning electron microscope (ESEM), equipped with a 500 µm aperture gaseous secondary electron detector and a Peltier cooling stage. Condensation and evaporation conditions in the ESEM were obtained by imaging at a constant sample temperature of 4 °C, while varying the water vapor pressure (RH). All environmental scanning electron microscopy experiments were carried out under the following operating conditions: beam accelerating voltage 10 kV, spot size 3.0, sample working distance 6.6 ± 1.3 mm, and 0.1–0.3 ms scanning speed. Secondary electron images were captured with a coupled device camera at 800×, 1600×, 3000×, and 6000× magnifications.

Particle swelling was observed by increasing the RH by 10% increments from 20% (1.2 Torr) to over 100% (>100% RH, 6.9 Torr). At each step, an image was taken after a swelling equilibration time of 5 min. The samples were maintained at >100% RH (6.4 and 6.9 Torr) for a total time of 10 min.

Images were analyzed for % swelling using GIMP, the GNU Image Manipulation Program (developed at University of Berkley, Berkeley, CA, USA). The brightness/contrast and colorize adjustment (hue 15–21; saturation 15–24; lightness −2) functions by GIMP software were applied for each micrograph. The increase in area was then calculated from the following equation (Equation (1)):*Swelling* (%) = ((*A_h_* − *A*_0_)/*A*_0_)) × 100 (1)
where *A_h_* represents the area (μm^2^) of the hydrated samples (>100% RH) and *A*_0_ represents the area (μm^2^) of the sample at 20% RH.

To observe the process of hydration and dehydration, MPC powder particles were imaged under three exposures. The first exposure was performed at 4 °C, 50% RH for 5 min, pre-hydration (microstructure characterization and RH equilibrium). The sample was allowed to equilibrate under these conditions for 5 min prior to increasing the RH. The second exposure was performed at 4 °C, >100% RH (continuous observation of hydration, liquid phase). Images were recorded at 6 min intervals for 30 min. The MPC particles were, then, maintained under these conditions for a further 60 min. The third exposure was performed at 4 °C, 50% RH, post-hydration (observation of residue and structural characteristics). The vapor pressure was decreased to its initial value of 3.0 Torr (50% RH), resulting in the evaporation of liquid water and allowing for the observation of the morphology of the resultant residue.

### 2.3. Initial Water Content (IWC) Determination

Powder samples (0.5–1.0 g) were dried at 60 °C at an absolute pressure (P_abs_) of <10 mbar for 24 h in a Jeio Tech OV-12 vacuum oven (Jeio Tech^®^, Seoul, Korea). The difference in powder weight before and after drying (g/100 g of dry solids) was defined as the initial water content (IWC) [25].

### 2.4. Differential Scanning Calorimetry (DSC)

Firstly, each powder was stored in evacuated desiccators (21 ± 2 °C) for 12 days over saturated solutions of LiCl, CH_3_COOK, MgCl_2_, and K_2_CO_3_ (Sigma Chemical Co., St. Louise, MO, USA), which, when equilibrium was reached, provided RH values of 11.4, 23.1, 33.2, and 44.1%, respectively, with a water activity (a_w_) value of 0.01 × % RH. Samples were weighed after 96 h and these values were taken as the equilibrium water contents. The a_w_ of each powder was measured using a Novasina, Labmaster.aw (Novatron, London, UK). The glass transition temperature (T_g_) of amorphous MPC powders stored at a_w_ values ranging from 0.11 to 0.44 were measured using a Q200 Differential scanning calorimetry (DSC, TA instruments^®^, New Castle, DE, USA). Each sample was transferred into preweighed standard DSC aluminium pan (40 μL, Tzero Hermetic Lid, TA instruments^®^, New Castle, DE, USA) and hermetically sealed. An empty pan was used as a reference. For anhydrous systems (0 a_w_), the lids of DSC aluminium pans were punctured to allow evaporation of residual water during the measurement. All samples were scanned from ~30 °C below to over the T_g_ region at a heating rate of 5 °C/min and subsequently cooled back to the initial temperature at 10 °C/min. Then, a second heating scan was run to 50 °C above the T_g_ at a heating rate of 5 °C/min. The onset of T_g_ was determined by the TA Universal Analysis software (TA instruments^®^, New Castle, DE, USA). For powders with a high-water content, the onset T_g_ was calculated using the Gordon–Taylor equation (Equation (2)):(2)$$T_{g}=\frac{w_{1}T_{g1}+{\mathrm{kw}}_{2}T_{g2}}{w_{1}+{\mathrm{kw}}_{2}}$$
where w_1_ and w_2_ are the mass fractions of amorphous material and water, respectively; T_g1_ and T_g2_ are the glass transition temperatures, respectively; k is a constant; and T_g2_ = −135 °C taken as the T_g_ of water [28].

### 2.5. Dynamical Mechanical Analyses (DMA)

The mechanical properties: E”, loss modulus (mechanical energy dissipation); E’, storage modulus (mechanical energy storage); and tan δ = E”/E’, of anhydrous and humidified MPC powders (described above for the DSC experiments) were measured using a Q800 Dynamical mechanical analyzer (TA instruments^®^, New Castle, DE, USA). The instrument was balanced and set at zero to determine the zero-displacement position before measurements commenced. Approximately 0.1 g of sample was placed on a metal pocket-forming sheet. This sheet was fixed inside the dual cantilever between the clamps. All results were obtained using the TA Universal Analysis software. Samples were scanned from ~50 °C below to over the α-relaxation region at a cooling rate of 5 °C/min and heating rate of 0.5 °C/min using the dual-cantilever bending mode. The α-relaxation temperatures (T_α_) were determined from frequency-dependent spectra of tan δ above the glass transition [18,19,29,30].

To calculate the relaxation times (τ) of peak T_α_, measured by DMA at various frequencies (f), Equation (3) was used [30,31]:(3)$$\tau =\frac{1}{2\pi f}$$

### 2.6. Rheology of MPC Dispersions

The MPC powders were dispersed (3 h stirring at 50 °C) in deionized water at 5, 10, 15, and 20%, dry matter (taking initial water content into consideration). Dispersions were agitated using a magnetic stirrer for 2 h. An ARES-G2 Rheometer ARES-G2 (TA Instruments^®^, New Castle, DE, USA) equipped with an aluminum parallel plate (40 mm) geometry was used to determine the apparent viscosity of solutions. A shear rate of 100 s^−1^ was used and maintained constant for all measurements across a temperature ramp from 5 to 60 °C. The viscosity profiles are shown in Appendix A. To convert the apparent viscosity (η) data into shear relaxation times (τ_s_), the Maxwell relation (Equation (4)) was used [28]:η = G_∞_τ_s_(4)
where G_∞_ is the infinite frequency shear modulus.

### 2.7. Calculation of the Williams–Landel–Ferry (WLF) Model Constants and Structural Strength Parameter

The constants C_1_ and C_2_ from the Williams–Landel–Ferry (WLF) equation were obtained, as described by Roos and Drusch (2015). The WLF equation in the form of Equation (5) was used to fit DMA and rheology data [21]:(5)$$\log_{10}\frac{\tau }{\tau_{s}}=\log_{10}\frac{\eta }{\eta_{s}}=\frac{-C_{1}\left(T-T_{g}\right)}{C_{2}+\left(T-T_{g}\right)}$$
where τ is the relaxation time, τ_s_ is the reference relaxation time, η is viscosity, η_s_ is reference viscosity, T is temperature, T_g_ is glass transition temperature, and C_1_ and C_2_ are constants.

The WLF equation in the form of Equation (6) suggested that the plot of 1/lg(τ/τ_s_) versus 1/(T − T_g_) gives a linear correlation:(6)$$\frac{1}{\mathrm{lg}\frac{\tau }{\tau_{s}}}=\frac{1}{-C_{1}}-\frac{C_{2}}{C_{1}\left(T-T_{g}\right)}$$

The WLF constants C_1_ and C_2_ were derived from the slope and interception [32].

Mathematically, structural strength is based on the WLF relationship and can be calculated by equation (Equation (7)):(7)$$S=\frac{{\mathrm{dC}}_{2}}{C_{1}-d}$$
where d is a parameter, showing the critical decrease in the number of logarithmic decades for the flow (e.g., from 100 s to 0.01 s corresponds to d = 4, which can be chosen for each system as an integer depending on the critical time for the process), C_1_ and C_2_ are “non-universal” constants in the WLF equation.

Equation (8) was used to predict structural strength at different water contents:(8)$$S=\frac{w_{1}S_{1}+{\mathrm{kw}}_{2}S_{2}}{w_{1}+{\mathrm{kw}}_{2}}$$
where w_1_ is the weight fraction of dry solid, w_2_ is the weight fraction of water, k is a coefficient, S_1_ is the structural strength for anhydrous system, S_2_ is the structural strength of pure water (S_2_ = 6.0) [19].

### 2.8. Data Analysis

Mean data of the IWC, DSC, and DMA were calculated from 3 replicates with standard deviations expressed in error bars. Statistical analysis was performed using a paired-sample *t*-test in Microsoft Office Excel 2011 (Microsoft, Inc., Redmond, WA, USA). Means differ significantly from each other if *p* < 0.05.

## 3. Results and Discussions

### 3.1. Environmental Scanning Electron Microscopy

The ESEM micrographs (Figure 1) show topographical differences between MPC powders. Regular (R) particles were dispersed with minimal face to face aggregation, whilst agglomeration produced clusters consisting of several particles. Nitrogen injection created more inflated and spherical powder particles as compared with R and agglomerated (A) controls, as a result of the increased occluded gas spaces; several NI powder particles exhibited a fractured and porous surface. The broken particles revealed the presence of tightly closed air bubbles below the particle’s surface (Figure 1, arrow) created by the injection and subsequent rapid removal of nitrogen; the spaces between the air bubbles likely caused the superficial pores. The MPC powder characteristics detected with ESEM were consistent with observations taken by a scanning electron microscope [15].

The ESEM surface analysis was performed to better understand water–particle interactions during swelling, liquid hydration and dehydration, and to elucidate the mechanisms of surface variations among the MPC powders. No apparent change in particle swelling was noted between the initial 20% RH (1.2 Torr) up to 40% RH (2.4 Torr). As the RH gradually increased from a value of 50% (3.0 Torr) to 90% (5.5 Torr), variation in particle swelling was apparent in all MPC particles. The AN- and RN-MPC particles exhibited more pronounced signs of swelling in clustered particles of various sizes. At 100% RH, the AN and RN particles swelled more rapidly, and the particles’ surfaces appeared fuller and rounder. As RH exceeded 100% (>100%, 6.9 Torr), the hydration further intensified with the presence of liquid water. A comparison of particle size between 20% RH (starting value) and >100% RH (final swelling) (Equation (1)) for each powder sample showed an increase in size of 25%, 20%, 40%, and 97% for R, A, AN, and RN, respectively. These results were in agreement with those of McSweeney et al., 2021b, who showed that NI promoted a faster uptake of moisture in the early stages of water sorption as compared with R and A controls [33].

The process of hydration and dehydration of MPC powder particles was carried out under three distinct environmental conditions: pre-hydration, hydration, and post-hydration (Figure 2 and Figure 3). Pre-hydration (Figure 2A, 50% RH), all MPC powder particles exhibited comparable morphology to the particles observed by ESEM during the previous swelling experiments. Initially, during the hydration (liquid phase) at >100% RH, the surface of all MPC powder particles began to fill out and become smoother (Figure 2B), as previously seen in other MPC80 powders [27]. With time, the particles started to fuse, although to a lesser degree in the case of R. Conformational changes could be seen in the A- and AN-MPC powder particles, the RN-MPC powder particles folded inwards, and the R-MPC powder particles displayed no significant conformational changes.

Post-hydration, the chamber vapor pressure was reduced to 50% RH to evaporate liquid water. The ESEM micrographs (Figure 3) depict the subsequent residue and morphological changes in MPC powder particles. The R-MPC powder particles exhibited the poorest solubility, with little surface erosion evident. The A-MPC powder particles showed a small degree of solubility, with some surface breakdown and erosion. A solid film was evident, indicating the particles’ partial dissolution. The solubility of AN powder particles was markedly improvement as compared with that of R- and A-MPC powder particles. The RN powder particles, after desorption, revealed particle shrinkage, breakdown, and collapse of the particles’ structure, resulting in a much smaller size and more compact network structure. The ESEM findings support the principle that high pressure N_2_ injection prior to spray during improves the solubility of MPC powders.

### 3.2. Water Content and Glass Transition Temperature of Anhydrous and Humidified Systems

Table 1 shows the initial water content (IWC) and water content of powders stored for 120 h at different relative humidity values (RHs). The water content increased with increasing RH, which was previously seen by McSweeney et al., 2021b [33]. Nitrogen injection had no significant effect on the IWC, with R- and RN-MPC powders having similar values, and A- and AN-MPC powder also similar. However, agglomerated powders had significantly lower IWC as compared with regular powders. The lower IWC of agglomerated MPC powders may be due to the additional drying step that the fine particles received, as they passed through the dryer more than once as compared with one pass for regular powders. This trend in water content was reversed when the powders were equilibrated at 11, 23, 33, and 44% RH, with agglomerated powders containing higher water content than their regular powder counterparts, due to their more porous particle structure which allowed increased moisture uptake [34,35]. It is important to note that the differences between powders is solely attributed to particle structure and drying conditions, as all powders contained the same chemical composition [33].

The onset of calorimetric T_g_ for anhydrous and humidified (0–44% RH) MPC systems are shown in Table 2. The results were calculated from the first heating step which provided a “fingerprint” of the sample [36]. At 0% RH, there were differences in T_g_ between MPC powders, with the R-MPC powder having the lowest T_g_ value (111 °C) and the AN-MPC powder having the highest system T_g_ value at 125 °C. Interestingly, there was a significant trend between regular and agglomerated powders, irrespective of NI, with agglomerated powders consistently having lower T_g_ values than regular powders at 23, 33, and 44% RH (Table 2). One previous study which looked at the T_g_ properties of foods systems with the same chemical composition was by Haque and Roos [37]. These authors found that freeze dried lactose had a higher T_g_ value as compared with spray dried lactose, although this difference seemed to be reduced when whey protein or gelatin was added. A study by Qiao et al. (2011) used finite element modeling showed that the T_g_ of agglomerated polymer nanocomposites was reduced as compared with non-clustered, more homogenous nanocomposites. The authors stated that agglomeration reduced T_g_ due a reduction in volume fraction [38]. However, the addition of water significantly decreased the T_g_ values due to the extremely high plasticization effect of water and its extremely low T_g_ value (−135 °C) [28]. Cohesiveness, as a property of the attraction forces between particles is extremely sensitive to water content [32,39]. Therefore, for humidified samples (RH 23–44%) R powders showed a higher T_g_ as compared with the more clustered A-MPC powders. This is in agreement with the water content in the systems as shown in Table 1. Within the MPC there are probably two main components responsible for the interactions occurring at the T_g_, i.e., protein and lactose. A high humidity may lead to increased levels of hydrogen bonding within the secondary structure of the casein polymers via interactions with polar amino acid side chains as well as increased plasticization of amorphous lactose. From a practical view point, there may be a relationship between the T_g_ and subsequent MPC powder functionality, as it is known that increased humidity and temperatures during the storage of MPC powders leads to a significant decrease in the rehydration and solubilization properties.

Based on the experimentally measured T_g_ data for MPC powders (Table 2), the Gordon–Taylor equation (Equation (2)) was used for the prediction of T_g_ across the water content range 0–100% (*w*/*w*) (Figure 4). Additionally, Appendix A shows the Gordon–Taylor predicted T_g_ values obtained from Figure 4 for MPC dispersions (5, 10, 15, and 20% total solids). The predicted T_g_ values of diluted systems are very low due to the extremely low T_g_ of water. Therefore, the values of T_g_ decreased with increasing water content for each MPC system, but with no significant difference observed between any of the MPC dispersions. This is to be expected, given that all powders were fully hydrated and solubilized prior to analysis. Using experimental T_g_ values (T_g1_, Table 2) for anhydrous (RH 0%) MPC powders and a T_g2_ of −135 °C [28] for pure water, the following constants (ks) for MPC systems were used: 8.0 ± 1.1 for regular control (R), 5.6 ± 1.0 for agglomerated control (A), 9.2 ± 0.8 for regular NI (NR), and 6.0 ± 1.0 for agglomerated NI (AN) powder.

### 3.3. Dynamic-Mechanical Properties

The mechanical α-relaxation temperatures (T_α_) for MPC powders are shown in Table 3. Significant changes occurred in the molecular mobility of powders with increasing temperature. This result was similar to previous studies which reported an increase in the loss moduli with increasing a_w_ in dairy-based systems, due to the plasticizing effect of water [29,40].The NI powders (RN and AN) showed slightly higher T_α_ values as compared with their non-NI counterparts (R and A). A similar effect was also seen for agglomerated powders, whereby, the T_α_ values were lower as compared with R-MPC and RN-MPC powders, and, as all powders were manufactured from the same starting material, the difference in T_α_ are due to powder particle structure and shape [15,37]. Therefore, the mode of drying and dryer configuration have significant knock-on effects on the mechanical properties of powders. In theory this may become even more pronounced when drying sticky type systems such as infant milk formulas, where high levels of lactose are often added. In the glassy state, material matrices have a high stability, as only molecular short-range vibrations (β–relaxations) are possible. This state is also characterized by extremely long relaxation times ~10^2^ s [28]. Structural α-relaxation is the large-scale molecular reorganization of a material and occurs in amorphous systems at temperatures ~20–30 °C above the onset to the calorimetric T_g_. This process decreases the structural relaxation time down to 10^−14^ s [41,42].

### 3.4. Williams–Landel–Ferry (WLF) Modeling and Structural Strength (S)

The temperature dependency of viscosity and structural relaxation time in food systems is conventionally described using the WLF equation [19,21]. The “non-universal” C_1_ and C_2_ WLF-model constants of anhydrous and humidified MPC powders are shown in Table 4. The calculation assumes that viscosity and relaxation times of the glassy state dramatically decrease (down to 10^5^ Pa s and 10^−14^ s, respectively) in the rubbery state at temperatures immediately above T_g_ [43,44]. A “strength” concept by Roos and co-authors, describes the flow characteristics of amorphous materials based on temperature dependence of structural relaxation times in different food systems in the vicinity of and above the T_g_ [18,19,20]. During heating, a critical and large change in the structural relaxation time occurred between 0.01 and 100 s, or between −2 and +2 in logarithmic scale (*d* = 4).

Strength may be defined as the temperature difference between T_g_ and T (here T = T_α_), at which the relaxation time is *d* times shorter than the T_g_. Figure 5 (data points) shows the structural strength (S) parameters (calculated using Equation (7)) for MPC powders across a range of RH values, and shows *d = 4* (i.e., −2 to +2 in logarithmic scale) for all systems. Extrapolating the experimental data, using the WLF constants (Table 4), shows a clear decrease in the S temperature for all powders, when changing RH from 0 to 23% (Figure 5). At 0% RH, non-NI powders (Figure 5A,B) showed that the S values became linear at temperatures >65 °C as compared with S values of ≤65 °C for RN-MPC and AN-MPC powders (Figure 5C,D, respectively). This indicated that the injection of nitrogen significantly reduced molecular strength under anhydrous conditions. Increasing the RH from 23 to 44% in non-NI powders (i.e., R-MPC and A-MPC) showed very little difference, with a S temperature of ~40 °C as compared with a significant decrease in S temperature for RN-MPC and AN-MPC powders (i.e., ~30 °C, Figure 5C,D). Appendix A shows the actual average S values for all powders. NI decreased S due to changes in the microstructure of NI powders and the fact that there was a significant increase in porosity of the RN-MPC powder particles, which made them more susceptible to breakage. These differences in S (~6 °C) occurred irrespective of powder composition, indicating that NI had a significant effect on the molecular mobility of MPC powders and, for non-NI systems, structural relaxation times achieved a critical level at higher temperatures than in NI systems.

## 4. Conclusions

In this study, the effect of nitrogen gas injection, prior to spray drying, on the glass transition temperature, structural relaxation, and strength, as well as rheology and real-time hydration of high protein regular and agglomerated dairy powders was investigated.

Overall, water content, as expected, had the most significant effect on the thermomechanical properties of dairy powders and modifying powder particle morphology had only a slight influence on T_g,_, indicating that it is the composition of powders that has the most dramatic effect on glass transition, although NI did result in significant differences in T_α_ and strength values between non-NI powders and their NI counterparts. The implications for altering powder particle morphology on the subsequent thermomechanical properties are fairly negligible, which is an important finding in that changes can be made to particle structure without causing significant changes to T_g_. However, the ESEM images confirmed that NI increased particle porosity, promoted a faster uptake of moisture, and enhanced solubility properties as compared with non-NI powders. The faster uptake of water at a given relative humidity may result in lower product stability and allow molecular interactions to occur at a lower temperature. These results provide useful information on how changing powder particle structure, as opposed to chemical composition which is often studied, caused a slight change in the thermo-dynamic properties of high protein powders. Possible future work could examine the effects of agglomeration and gas injection on the potential changes in high lactose systems, such as infant milk formula.

## Figures and Tables

**Figure 1 foods-11-00292-f001:**
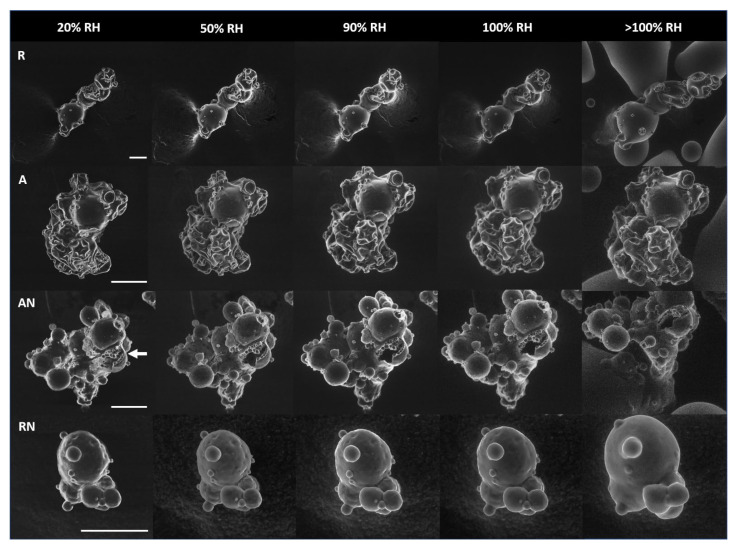
ESEM micrographs of MPC particles’ swelling captured at 800×, 1600× and 3000× magnifications. The rows show the hydration of regular (R), agglomerated (A), agglomerated NI (AN), and regular NI (RN) MPC powder particles, at 4 °C. The columns show the particles at 20% RH (1.2 Torr), 50% RH (3.0 Torr), 90% RH (5.5 Torr), 100% RH (6.1 Torr), and >100% RH (6.9 Torr). Scale bars represent 50 µm. The arrow clearly shows the occluded air bubbles below the particle’s surface created by nitrogen injection in the fractured AN powder particle.

**Figure 2 foods-11-00292-f002:**
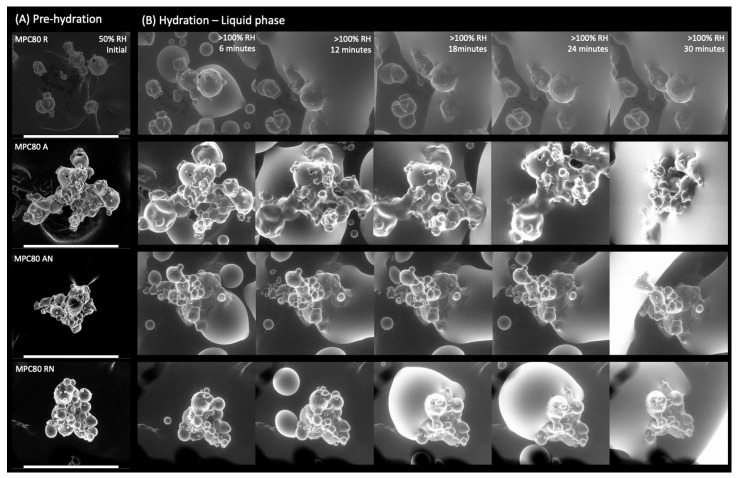
ESEM micrographs showing the dynamic process of hydration (rows) captured at 800× magnification of regular (R), agglomerated (A), agglomerated NI (AN), and regular NI (RN) milk protein concentrate powder (MPC) particles at 4 °C: (**A**) MPC particles observed prior to hydration at 50% RH (3.0 Torr); (**B**) MPC particles during hydration at >100% RH (6.4 Torr) at 6, 12, 18, 24, and 30 min. Scale bars represent 300 µm.

**Figure 3 foods-11-00292-f003:**
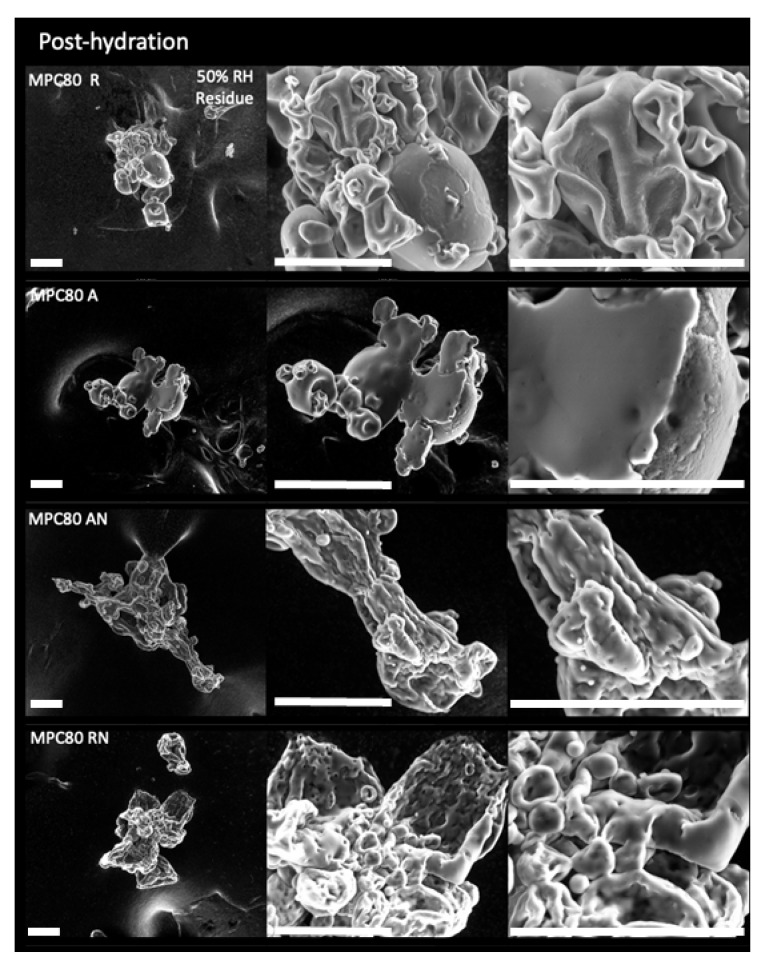
ESEM micrographs showing the post-hydration of the samples in Figure 2 for regular (R), agglomerated (A), agglomerated NI (AN), and regular NI (RN) milk protein concentrate powder (MPC) particles at 4 °C. Columns show the morphology of the resultant residue captured at 800×, 3000×, and 6000× magnifications. Scale bars represent 50 µm.

**Figure 4 foods-11-00292-f004:**
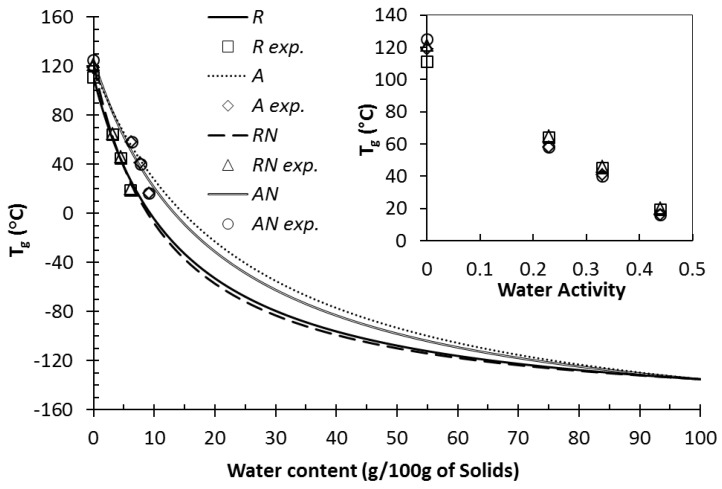
Glass transition temperature (T_g_) measured as a function of water content and water activity (a_w_) of regular (R), agglomerated (A), regular nitrogen-injected (RN), and agglomerated nitrogen-injected (AN) milk protein concentrate powders (MPCs). Lines correspond to the T_g_ predicted by the Gordon–Taylor equation. Symbols correspond to T_g_ values obtained experimentally by differential scanning calorimetry.

**Figure 5 foods-11-00292-f005:**
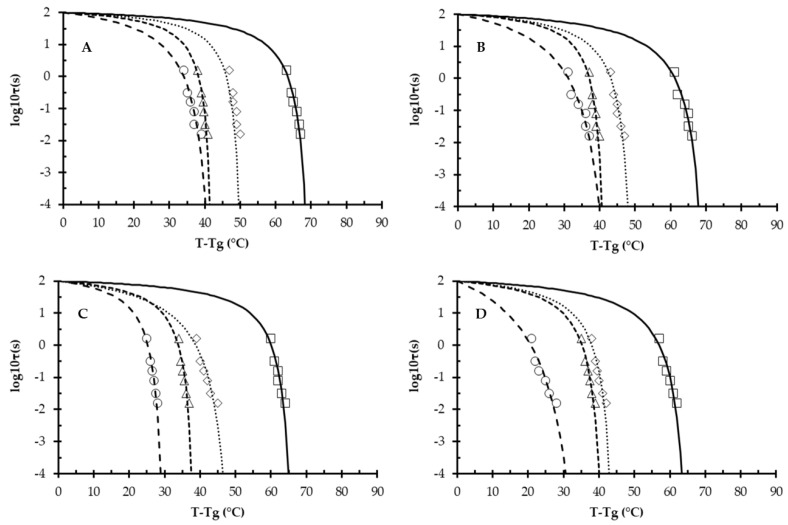
Modified Williams–Landel–Ferry curves (lines) and experimental data (symbols) for: (**A**) regular; (**B**) agglomerated; (**C**) regular nitrogen-injected; (**D**) agglomerated nitrogen-injected milk protein concentrate powders, measured as a function of relative humidity at 0 (□), 23 (◊), 33 (∆), and 44% (ο).

**Table 1 foods-11-00292-t001:** Initial water content (IWC) and equilibrated water content values of regular (R), regular NI (RN), agglomerated (A), and agglomerated NI (AN) MPC systems stored for 120 h at different relative humidity (RH) values, at 21 ± 2 °C.

RH %	Water Content (g/100 g of Dry Solids)			
	R	A	RN	AN
IWC	5.37	4.96	5.59	5.16
11%	2.4 ± 0.58	4.8 ± 0.26 *	2.7 ± 0.26	5.0 ± 0.37 *
23%	3.2 ± 0.01	6.2 ± 0.10 *	3.2 ± 0.06	6.3 ± 0.12 *
33%	4.6 ± 0.02	7.7 ± 0.14 *	4.5 ± 0.02	7.8 ± 0.15 *
44%	6.1 ± 0.05	9.1 ± 0.06 *	6. 0 ± 0.01	9.2 ± 0.21 *

IWC, water content of powders obtained directly after drying. Values presented are the means of triplicate measurements ± standard deviation. * Denotes values of powders significantly different to the control (i.e., R-MPC) powder.

**Table 2 foods-11-00292-t002:** Measured by differential scanning calorimetry (DSC), the onset of calorimetric glass transition temperature (T_g_) of regular (R), regular NI (RN), agglomerated (A), and agglomerated NI (AN) MPC powders stored for 120 h at 21 ± 2 °C at different relative humidity values (RHs).

RH %	Glass Transition Temperature (T_g_) of MPC Systems, °C			
	R	A	RN	AN
0	111 ± 5	119 ± 3 *	121 ± 3 *	125 ± 5 *
23	64 ± 3	58 ± 3 *	65 ± 2	58 ± 3 *
33	45 ± 2	41 ± 5 *	46 ± 1	40 ± 4 *
44	19 ± 4	16 ± 3 *	20 ± 4	16 ± 3 *

Values presented are the means of triplicate measurements ± standard deviation. * Denotes values of powders significantly different to the control (i.e., R-MPC) powder.

**Table 3 foods-11-00292-t003:** The α-relaxation temperature (T_α_) values obtained by dynamic mechanical analysis (DMA) of regular (R), regular NI (RN), agglomerated (A), and agglomerated NI (AN) MPC powders equilibrated at different relative humidity values (RH) and measured at different frequencies (f) and relaxation times (logτ, s).

**RH 0%**		**R**	**A**	**RN**	**AN**
**f, Hz**	**logτ, s**	**T_α_, °C**	**T_α_, °C**	**T_α_, °C**	**T_α_, °C**
0.1	0.20	174 ± 7	180 ± 11 *	181 ± 11 *	182 ± 9 *
0.5	−0.49	176 ± 5	181 ± 7 *	183 ± 9 *	184 ± 11 *
1.0	−0.80	177 ± 6	183 ± 8 *	184 ± 9 *	185 ± 9 *
5.0	−1.50	178 ± 7	184 ± 5 *	184 ± 7 *	186 ± 9 *
10.0	−1.80	178 ± 6	185 ± 11 *	185 ± 12 *	187 ± 10 *
**RH 23%**		**R**	**A**	**RN**	**AN**
**f, Hz**	**logτ, s**	**T_α_, °C**	**T_α_, °C**	**T_α_, °C**	**T_α_, °C**
0.1	0.20	111 ± 5	101 ± 7 *	104 ± 7	96 ± 4 *
0.5	−0.49	112 ± 4	102 ± 6 *	105 ± 7 *	97 ± 7 *
1.0	−0.80	112 ± 6	102 ± 5 *	107 ± 6	98 ± 5 *
5.0	−1.50	113 ± 5	104 ± 4 *	108 ± 6	99 ± 7 *
10.0	−1.80	114 ± 7	105 ± 5 *	110 ± 7	100 ± 9 *
**RH 33%**		**R**	**A**	**RN**	**AN**
**f, Hz**	**logτ, s**	**T_α_, °C**	**T_α_, °C**	**T_α_, °C**	**T_α_, °C**
0.1	0.20	83 ± 5	78 ± 4 *	80 ± 4	75 ± 5 *
0.5	−0.49	84 ± 4	79 ± 6 *	81 ± 5	76 ± 4 *
1.0	−0.80	85 ± 5	79 ± 4 *	82 ± 4	78 ± 5 *
5.0	−1.50	85 ± 6	80 ± 5 *	82 ± 3	78 ± 6 *
10.0	−1.80	86 ± 5	81 ± 5 *	83 ± 5	79 ± 5 *
**RH 44%**		**R**	**A**	**RN**	**AN**
**f, Hz**	**logτ, s**	**T_α_, °C**	**T_α_, °C**	**T_α_, °C**	**T_α_, °C**
0.1	0.20	53 ± 3	47 ± 5 *	45 ± 3 *	37 ± 4 *
0.5	−0.49	54 ± 5	48 ± 3 *	46 ± 2 *	38 ± 5 *
1.0	−0.80	55 ± 4	50 ± 4	47 ± 4 *	39 ± 3 *
5.0	−1.50	56 ± 5	52 ± 4	47 ± 3 *	42 ± 3 *
10.0	−1.80	58 ± 4	53 ± 5	48 ± 2 *	44 ± 4 *

Values presented are the means of triplicate T_α_ measurements ± standard deviation. * Denotes values of powders significantly different to the control (i.e., R-MPC) powder.

**Table 4 foods-11-00292-t004:** Calculated Williams–Landel–Ferry (WLF) constants C_1_ and C_2_ for regular (R), regular NI (RN), agglomerated (A), and agglomerated NI (AN) MPC powders equilibrated at different relative humidity (RHs).

System	RH 0%		RH 23%		RH 33%		RH 44%	
	-C_1_, s	-C_2_, °C	-C_1_, s	-C_2_, °C	-C_1_, s	-C_2_, °C	-C_1_, s	-C_2_, °C
R	0.2 ± 0.06	71.0 ± 3.2	0.2 ± 0.06	52.6 ± 2.7 *	0.3 ± 0.09	43.5 ± 3.3 *	0.6 ± 0.1 *	44.1 ± 2.1 *
A	0.3 ± 0.05	71.8 ± 3.7	0.4 ± 0.09	50.8 ± 2.5 *	0.3 ± 0.11	42.8 ± 2.3 *	1.0 ± 0.3 *	46.5 ± 2.9 *
RN	0.2 ± 0.05	67.6 ± 3.7	0.7 ± 0.08 *	51.5 ± 2.2 *	0.4 ± 0.09	39.9 ± 3.6 *	0.5 ± 0.09 *	31.5 ± 3.4 *
AN	0.4 ± 0.09	67.2 ± 5.3	0.4 ± 0.08	45.9 ± 2.7 *	0.4 ± 0.10	43.0 ± 2.4 *	1.9 ± 0.2 *	40.2 ± 2.4 *

C_1_ and C_2_ constant values presented are the means of triplicate measurements ± standard deviation. * Denotes values of powders significantly different to the control (i.e., R-MPC) powder.

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
