# Peer review of "The Effect of High Protein Powder Structure on Hydration, Glass Transition, Water Sorption, and Thermomechanical Properties"

_foods, 2022, doi:10.3390/foods11030292_

Round 1

Reviewer 1 Report

The article entitled "The effect of high protein powder structure on hydration, glass transition, water surprise and thermomechanical properties has been evaluated." The subject of the article is interesting and the experimental development has been extensive and very well explained. There are minor observations regarding explaining the water-food interaction a little more thoroughly. I believe that there are some aspects of the methodologies that should be explained in more abundance. I suggest a graphical abstrac of the experimental process for the readers' understanding. But in general terms the results are interesting and detailed.

Reviewer 2 Report

This is an interesting and valuable manuscript describing the effect of nitrogen gas injection, prior to spray-drying, on the glass transition temperature, structural relaxation, and strength as well as rheology and real-time hydration of high protein regular and agglomerated dairy powders. But the following points need to be done by the authors:

The abstract should be more informative by giving real results rather than elastic sentences.

The authors should mention the reason for selecting MPC instead of MPI clearly.

Tables 1, 2, 3, 4, 5, 6, and 7: Please provide statistical lettering.

Section 3.5: Why did not you do the rheology test for all the treatments? Would not it provide better and more useful results for analyzing the behavior of the article treatments?

Enrich discussion with recent references of not older than 5 years.

Conclusion: what is the future of your findings? The conclusion is not insightful, what are suggestions?

Round 2

Reviewer 2 Report

It could be accepted for publishing.